

# Quantification of the effect of modeled lightning $NO_2$ on UV-visible air mass factors

Joshua L. Laughner[1] and Ronald C. Cohen[1,2]

[1]Department of Chemistry, University of California, Berkeley, Berkeley, CA 94720
[2]Department of Earth and Planetary Sciences, University of California, Berkeley, Berkeley, CA 94720

*Correspondence to:* R.C. Cohen (rccohen@berkeley.edu)

**Abstract.** Space-borne measurements of tropospheric nitrogen dioxide ($NO_2$) columns are up to 10x more sensitive to upper tropospheric $NO_2$ than near-surface $NO_2$ over low reflectivity surfaces. Here, we quantify the effect of adding simulated lightning $NO_2$ to the a priori profiles for $NO_2$ observations from the Ozone Monitoring Instrument using modeled $NO_2$ profiles from the Weather Research and Forecasting—Chemistry (WRF-Chem) model. With observed $NO_2$ profiles from the Deep Convective Clouds and Chemistry (DC3) aircraft campaign as observational truth, we quantify the bias in the $NO_2$ column that occurs when lightning $NO_2$ is not accounted for in the a priori profiles. Focusing on the central and eastern United States, we find that a simulation without lightning $NO_2$ underestimates the air mass factor (AMF) by 26% on average for common summer OMI viewing geometry, and 35% for viewing geometries that will be encountered by geostationary satellites. Using a simulation with 500 to 665 mol NO flash$^{-1}$ produces good agreement with observed $NO_2$ profiles and reduces the bias in the AMF to $< \pm 4\%$ for OMI viewing geometries. The bias is regionally dependent, with the strongest effects in the southeast United States (up to 80%) and negligible effects in the central US. We also find that constraining WRF meteorology to a reanalysis dataset reduces lightning flash counts by a factor of 2 compared to an unconstrained run, most likely due to changes in the simulated water vapor profile.

## 1   Introduction

$NO_x$ ($\equiv NO + NO_2$) is a short-lived (typical summer lifetime 2–7 h) trace gas in the atmosphere. $NO_x$ is emitted by both anthropogenic and natural processes; the former is primarily due to combustion, while the latter includes biomass burning, soil bacteria nitrification or denitrification, and lightning. $NO_x$ regulates ozone production throughout the troposphere; therefore accurate measurements of $NO_x$ and understanding of $NO_x$ chemistry is essential to describe and predict the role of ozone as both an air quality hazard, oxidant, and a greenhouse gas.

Space-borne measurements of $NO_2$ as an indicator of total $NO_x$, such as those from the Global Ozone Monitoring Experiment (GOME and GOME-2), SCanning Imaging Absorption SpectroMeter for Atmospheric CHartographY (SCIAMACHY), and Ozone Monitoring Instrument (OMI) are a valuable tool in understanding $NO_x$ emissions and chemistry because of their global reach and long data records. Use of these observations includes assessment of $NO_x$ chemistry (e.g. Beirle et al., 2011; Valin et al., 2013) anthropogenic emissions (e.g. Miyazaki et al., 2012; Russell et al., 2012; Lu et al., 2015; Liu et al., 2016,





2017) and natural emissions (e.g. Martin et al., 2007; Beirle et al., 2011; Hudman et al., 2012; Mebust et al., 2011; Mebust and Cohen, 2013, 2014; Miyazaki et al., 2014; Zörner et al., 2016).

Retrieval of tropospheric $NO_2$ from a UV-visible satellite spectrometer requires three main steps: fitting the measured absorbance to produce a slant column density (SCD), separation of the stratospheric and tropospheric signals, and conversion of the tropospheric SCD to a vertical column density (Boersma et al., 2011; Bucsela et al., 2013). This final step accounts for the effect of variable path length through the atmosphere, surface elevation and reflectance, and the vertical distribution of $NO_2$ (Palmer et al., 2001). For observations over low reflectivity surfaces, the sensitivity of the satellite to $NO_2$ decreases towards the surface, as photons penetrating into the lower atmosphere may scatter into the surface, where most are absorbed; thus, there is a higher probability that a photon that reaches the detector has interacted only with the higher levels of the atmosphere (Hudson et al., 1995; Richter and Wagner, 2011). That is to say, a given number of $NO_2$ molecules in the upper troposphere produce a greater signal than the same number of $NO_2$ molecules at the surface would. Thus, a priori knowledge of the vertical profile of $NO_2$ is necessary to account for this effect in the retrieval.

These vertical profiles are simulated using chemical transport models (CTMs) such as TM4 (used in Boersma et al., 2011), the Global Modeling Initiative CTM (used in Bucsela et al., 2013), or the Weather Research and Forecasting—Chemistry (WRF-Chem, used in Russell et al., 2011). These models must account for atmospheric transport, chemistry, emissions, and deposition to accurately simulate the required $NO_2$ profiles. Most emission of $NO_2$ occurs at or very near the surface. There are comparatively weaker sources of $NO_2$ in the upper troposphere, limited to transport from the surface, aircraft, stratospheric mixing, and lightning (Jaeglé et al., 1998).

Simulation of lightning $NO_x$ emission in these models is typically done by assuming each flash emits a set number of molecules of NO. The number and location of lightning flashes is often parameterized using the method of Price and Rind (1992), which relates lightning flash rates to cloud top heights, which in turn are calculated from the model's meteorology. In CTMs focused on simulating surface chemistry to understand or predict air quality, such as WRF-Chem or the Community Multi-scale Air Quality (CMAQ) model, including $NO_x$ produced by lightning may be disabled by default or require the user to prepare additional input files. As these models are often used to simulate high resolution a priori profiles (e.g. Russell et al., 2011, 2012; Kuhlmann et al., 2015; Laughner et al., 2016; Goldberg et al., 2017), the absence of lightning $NO_x$ from the a priori profiles may contribute to a significant bias in the interpretation of the measurements.

In the upper troposphere, the lifetime of $NO_x$ is short near thunderstorms ($\sim 3$ h Nault et al., 2016), but longer (12–48 h) away from thunderstorms (Bertram et al., 2007; Apel et al., 2012). As a result, lightning $NO_x$ can affect upper tropospheric $NO_x$ concentrations distant from active storms; thus, simulated lightning $NO_x$ will have wide-reaching and persistent effects on a priori $NO_2$ profiles throughout a model domain. Previous work by, e.g. Beirle et al. (2009) and Pickering et al. (2016), has provided careful analysis of the effect of lightning on AMFs in the near field of a thunderstorm, with the goal of improving direct satellite measurements of the mean production of NO per flash. Our goal here is to consider the broader impact of modeled lightning $NO_x$ on satellite retrievals on the full domain both near and far from the lightning event.

In this work, we evaluate the impact of modeled lightning $NO_x$ on $NO_2$ a priori profiles simulated with the WRF-Chem chemical transport model for a domain covering the central and eastern US. We first consider the problem in a general sense,





with a sensitivity test using three profiles simulated with different amounts of lightning $NO_x$. We then compare modeled profiles to observations from the Deep Convective Clouds and Chemistry (DC3) campaign to determine the accuracy of AMFs derived using the simulated profiles, and finally implement these profiles in an $NO_2$ retrieval to demonstrate the spatial pattern and significance of this effect in a real application.

## 2 Methods

### 2.1 The Deep Convective Clouds and Chemistry Campaign

The Deep Convection Clouds and Chemistry (DC3) Campaign is an aircraft measurement campaign that took place between 18 May and 22 June 2012 throughout the central and southeastern US (Barth et al., 2015). The NASA DC-8 aircraft sampled outflow from convective systems, studying direct and aged lightning $NO_x$ emissions. We use $NO_2$ measurements made by laser induced florescence at 1 second resolution in this study (Thornton et al., 2000; Nault et al., 2015).

### 2.2 Weather Research and Forecasting—Chemistry Model

We use the Weather Research and Forecasting—Chemistry (WRF-Chem) model v. 3.5.1 (Grell et al., 2005) to simulate $NO_2$ profiles across a domain that covers the same region as the DC3 campaign at 12 km model resolution. Meteorological initial and boundary conditions are driven by the North American Regional Reanalysis (NARR) dataset. Chemical initial and boundary conditions are driven by output from the Model for Ozone and Related chemical Tracers (MOZART; Emmons et al., 2010) provided by the National Center for Atmospheric Research. Anthropogenic emissions are driven by the National Emissions Inventory 2011 (NEI 11); each emitted species is scaled domain-wide by the ratio of its total annual 2012 to 2011 emissions provided by the Environmental Protection Agency (EPA, 2016), e.g. 2012 $NO_x$ emissions are given at 13.657 million tons, 94% of the 2011 value of 14.519 million tons; the gridded 2011 NO emissions are multiplied by 0.94 to obtain 2012 emissions. Biogenic emissions are driven by the Model of Emissions of Gases and Aerosol from Nature (MEGAN; Guenther et al., 2006). The chemical mechanism is a customized version of The Regional Atmospheric Chemistry Model, version 2 (RACM2; Goliff et al., 2013) that includes updates to alkyl nitrate chemistry from Browne et al. (2014) and Schwantes et al. (2015), as well as formation, dissociation and photolysis of methylperoxy nitrate (Browne et al., 2011, see also http://wiki.seas.harvard.edu/geos-chem/images/GEOS_changes_MPN_chemistry.pdf) Instantaneous values of the model output are sampled every half hour.

WRF can be run such that the meteorology within the domain is driven by the model physics chosen, constrained by reanalysis meteorology data only through the initial and boundary conditions. Alternatively, four dimensional data analysis (FDDA) nudging (Liu et al., 2006) can be used to nudge the model meteorology towards a reanalysis meteorology product throughout the domain. We use this capability in two WRF-Chem simulations, nudging towards the NARR meteorology. In all other simulations, the meteorology evolves according to the model physics.





Lightning $NO_x$ emissions are calculated by the standard modules in WRF-Chem 3.5.1, with a slight modification to the assumed emission profile (described below). The flash rates (number of lightning flashes per unit time) are determined by the Price and Rind level of neutral buoyancy parameterization (Price and Rind, 1992), which depends on cloud top height, calculated using the Grell 3D cumulus physics (Grell, 1993; Grell and Dévényi, 2002) with Lin microphysics (Lin et al.,

1983). This number of flashes calculated may be scaled by a constant factor, we use this functionality for one run in Sect. 3.2, otherwise the scaling factor is 1. The intra-cloud/cloud-to-ground ratio is prescribed using the Boccippio et al. (2001) climatology; both intra-cloud and cloud-to-ground flashes are specified to generate the same number of mol NO per flash (Cooray et al., 2009; Ott et al., 2010), which for this study is 0, 500, or 665 mol flash$^{-1}$. These values are chosen to represent no lightning, the standard midlatitude assumption (500 mol flash$^{-1}$ Hudman et al., 2007), and the recently proposed 33%

increase in lightning $NO_x$ emissions of Nault et al. (submitted) (665 mol flash$^{-1}$).

The vertical distribution of NO emissions is driven by a modified version of the profiles from Ott et al. (2010). Several recent studies (Allen et al., 2012; Seltzer et al., 2015) suggest that the standard Ott profiles place too much $NO_x$ in the mid-troposphere. Ott et al. (2010) calculated these profiles using a polynomial fit to profiles of the post-convection vertical distribution of lightning $NO_x$ simulated by a cloud resolving model. The midlatitude profile generated by the cloud resolving

model has a bimodal distribution not captured by the polynomial fit; therefore we replace the standard (polynomial fit) Ott et al. (2010) midlatitude profile in WRF-Chem with the bimodal profile.

### 2.3 Matching aircraft and model data

We match WRF-Chem data to DC3 observations to evaluate the accuracy of the chosen lightning parameterization. Each 1 second DC3 $NO_2$ observation is paired with the corresponding WRF-Chem data point. Data points are matched in time by

finding the WRF-Chem output file (available every half-hour) nearest in time to a given DC3 observation.

Horizontally, a WRF-Chem data point is said to match with a DC3 observation if the latitude and longitude of the DC3 observation lie within the box defined by the midpoints of the WRF-Chem grid cell edges. These midpoints are computed as the average of the relevant corner coordinates (e.g. the western edge point is the average of the northwestern and southwestern corners); the corner coordinates are calculated by assuming that corners not on the edge of the domain are the average of the

four surrounding centers. Corners on the domain edge are calculated by extrapolating from the internal corners.

Vertically, we find the matching WRF-Chem data point from the column of such points identified by the previous two steps by finding the WRF-Chem grid point with the smallest difference in pressure compared to the DC3 observation. The result is two vectors of $NO_2$ concentrations (DC3 and WRF-Chem) that are the same length; WRF-Chem data points that correspond to multiple DC3 observations are repeated, thus inherently giving them more weight and reflecting the sampling of the aircraft.

### 2.4 The Ozone Monitoring Instrument

The Ozone Monitoring Instrument is a polar-orbiting, nadir-viewing UV-visible spectrometer on board the Aura satellite, launched in 2004. It has a nadir pixel size of $13 \times 24$ km$^2$. The primary detector is a 2D CCD array that observes a swath width of 2600 km and a spectral range of 270–500 nm (Levelt et al., 2006). It provides daily global observation for the first three





years of operation; after 2007 several detector rows developed anomalous radiances (termed the "row anomaly", http://projects.knmi.nl/omi/research/product/rowanomaly-background.php) that have expanded over time; from July 2011 on, this affects approximately one-third of the pixels. There are two publicly available global products of $NO_2$ column densities, the KNMI DOMINO product (Boersma et al., 2011) and the NASA Standard Product v3 (Krotkov et al., 2017), and numerous regional products, including OMI-EC (McLinden et al., 2014), Hong Kong OMI $NO_2$ (Kuhlmann et al., 2015), Peking University OMI $NO_2$ (POMINO Lin et al., 2015), Empa OMI $NO_2$ (EOMINO, http://temis.empa.ch/index.php), DOMINO2_GC (Vinken et al., 2014), and the Berkeley High Resolution OMI $NO_2$ retrieval (Russell et al., 2011, 2012).

## 2.5 Berkeley High Resolution OMI $NO_2$ retrieval

### 2.5.1 Retrieval product

To demonstrate the impact of modeled lightning $NO_x$ on retrieved $NO_2$ column densities, we use v2.1C of the BErkeley High Resolution (BEHR) $NO_2$ retrieval. Details of the algorithm are given in Russell et al. (2011); more recent updates are given in the changelog (http://behr.cchem.berkeley.edu/Portals/2/Changelog.txt). This product is available for download at http://behr.cchem.berkeley.edu/DownloadBEHRData.aspx.

Version 2.1C of the BEHR product is based on the NASA Standard Product version 2 (SP v2). It uses the OMI total slant column densities (SCDs) from the OMI $NO_2$ product OMNO2A v1.2.3 (Boersma et al., 2002; Bucsela et al., 2006, 2013), as well as the stratospheric separation and destriping from the NASA Standard Product v2.

The BEHR product recalculates the tropospheric air mass factor (AMF) using the formulation in Palmer et al. (2001). In previous versions of BEHR, the tropospheric AMFs and resulting vertical column densities (VCDs) were always "total" tropospheric columns, i.e., they included an estimated ghost $NO_2$ column below clouds. Starting in v2.1C, "visible-only" tropospheric AMFs and VCDs are included (which do not include the below-cloud ghost column), in addition to the "total" tropospheric VCDs. In both cases, separate AMFs for clear and cloudy scenes are calculated using Eq. (1)

$$\text{AMF} = \int_{p_0}^{p_{\text{tp}}} w(p) S(p) \, dp \tag{1}$$

where $p_0$ is the surface or cloud pressure (for clear and cloudy scenes, respectively), $p_{\text{tp}}$ is the tropopause pressure (fixed at 200 hPa), $w(p)$ are the pressure-dependent scattering weights from the TOMRAD look-up table used in the NASA SP v2 (Bucsela et al., 2013), which must be corrected for the temperature dependence of the $NO_2$ cross section:

$$w(p) = w_0(p) \left[ 1 - 0.003(T(p) - 220) \right] \tag{2}$$

where $w_0(p)$ is the scattering weight from the look-up table and $T$ is the temperature in Kelvin for a given latitude, longitude, and month; $T$ is taken from the same temperature profiles used in the NASA SP v2 (Bucsela et al., 2013).





Finally, $S(p)$, the shape factor, is computed as:

$$S(p) = \left( \int\limits_{p_s}^{p_{\mathrm{tp}}} g(p)\,dp \right)^{-1} g(p) \tag{3}$$

where $g(p)$ is the $NO_2$ vertical profile, and $p_s$ is either the surface or cloud pressure, depending on whether a total or visible-only tropospheric VCD is desired. BEHR v2.1C provides both. For clear scenes, $p_s$ is always the surface pressure. For cloudy

scenes, $p_s$ is the surface pressure when calculating the total VCD and the cloud pressure when calculating the visible-only VCD.

The clear and cloudy AMFs for a given pixel are combined as:

$$\mathrm{AMF_{total}} = (1 - f)\mathrm{AMF_{clear}} + f\,\mathrm{AMF_{cloudy}} \tag{4}$$

where $f$ is the radiance cloud fraction, i.e. the fraction of light from the pixel that is reflected off of clouds. The final VCD

is computed as:

$$\mathrm{VCD} = \frac{\mathrm{SCD}}{\mathrm{AMF_{total}}} \tag{5}$$

where the SCD is the tropospheric slant column density from the NASA SP v2.

The vector of scattering weights, $w(p)$, chosen from the TOMRAD look-up table depends on five parameters: solar zenith angle (SZA), viewing zenith angle (VZA), relative azimuth angle (RAA), albedo, and surface pressure. The SZA, VZA, and

RAA are directly provided or can be calculated from data provided in the NASA SP v2. The surface albedo for a given pixel is calculated by averaging the black sky albedo product MCD43C3 (Schaaf and Wang, 2015) values that fall within the pixel. This product is generated by the Moderate Resolution Imaging Spectroradiometer (MODIS) instruments on board the Aqua and Terra satellites. Clouds are assumed to have an albedo of 0.8 (Stammes et al., 2008). Surface pressures are calculated by averaging elevation data from the Global Land One-km Base Elevation project (Hastings and Dunbar, 1999) that falls within

the pixel and assuming a scale height of 7.4 km; cloud pressures are from the OMI O2-O2 algorithm (Acarreta et al., 2004; Sneep et al., 2008; Bucsela et al., 2013) and are included in the NASA SP v2.

When averaging over time for the results in Sect. 3.3 we only use pixels with the OMI cloud fraction $< 0.2$, XTrackQualityFlags $= 0$, and an even integer for VcdQualityFlags. The averages weight each pixel's contribution by the inverse of the pixel area. Unless otherwise stated, all results in this work use the total tropospheric column.

## 2.5.2 AMF sensitivity tests

To understand the sensitivity of the AMF to the profile shape under different conditions, we carry out sensitivity tests by varying the five input parameters to the TOMRAD look-up table. Table 1 lists the input parameters and the values used for



| Parameter | Abbreviation | Values | Unit |
|---|---|---|---|
| Solar Zenith Angle | SZA | 0, 11, 22, 33, 44, 55, 66, 77, 88 | deg. |
| Viewing Zenith Angle | VZA | 0, 14, 28, 42, 56, 70 | deg. |
| Relative Azimuth Angle | RAA | 0, 45, 90, 135, 180 | deg. |
| Albedo (clear sky) | Alb | 0, 0.009, 0.018, 0.027, 0.036, 0.044, 0.053, 0.062, 0.071, 0.080 | unitless |
| Albedo (cloudy sky) | Alb | 0.700, 0.722, 0.744, 0.767, 0.789, 0.811, 0.833, 0.856, 0.878, 0.900 | unitless |
| Surface Pressure (clear) | Surf P | 1013, 989, 965, 940, 916, 892, 868, 843, 819, 795 | hPa |
| Surface Pressure (cloudy) | Surf P | 1003, 930, 857, 783, 710, 637, 564, 490, 417, 344 | hPa |

**Table 1.** The values used for the five input parameters to the AMF TOMRAD lookup table in the sensitivity tests. Albedo and surface pressure have different sets of values when the sensitivity test is looking at clear sky and cloudy sky scenarios.

each parameter. For albedo and surface pressure, two sets of values are used; one represents common values seen for clear (unclouded) scenes, the other cloudy scenes. The range of values for SZA, VZA, and RAA span the values defined in the TOMRAD look-up table. The range of values for Albedo (clear sky), Surface Pressure (clear sky), and Surface Pressure (cloudy) span the average 5th and 95th percentiles of those values observed in seven days of BEHR data (2012-06-01 to 2012-06-07). The limits for Albedo (cloudy) are chosen as $0.8 \pm 0.1$, i.e. the assumed cloud albedo plus a reasonable range to explore.

Scattering weights are calculated for every combination of clear or cloudy parameters (27000 combinations). We choose the temperature correction (Sect. 2.5.1, Eq. 2) assuming the June temperature profile at $37.5° \, N$, $95° \, W$. Using a single $NO_2$ profile, an AMF is calculated for every combination of input parameters.

We use three types of $NO_2$ vertical profiles for the AMF sensitivity tests.

1. One derived from the 1 sec DC3 $NO_2$ data (Sect. 2.1)

2. One using WRF-Chem output matched to the DC3 flight path (Sect. 2.3)

3. One using WRF-Chem output averaged over the entire domain between 1700 and 2200 UTC (roughly the times during which OMI is over North America)

In all cases the data points (modeled or measured) used to generate the $NO_2$ profiles are binned by pressure to generate a profile defined at the same pressures (using pressure as a vertical coordinate) as the scattering weights in the look-up table. Each data point is placed in the bin with the scattering weight pressure closest to the pressure of the data point.





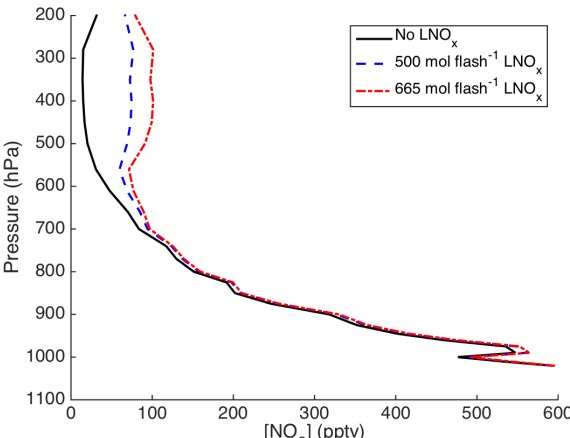

**Figure 1.** Domain-wide mean WRF-Chem $NO_2$ profiles.

## 3   Results

### 3.1   Parameter sensitivity study using modeled profiles

We begin by demonstrating the sensitivity of the AMF to modeled lightning $NO_x$ emissions in a general sense. Profiles used in this section are those derived by binning WRF-Chem output from the entire domain for simulations with 0, 500, and 665

mol NO flash$^{-1}$ without FDDA nudging (Fig. 1). Figure 2 shows the percent difference in the AMF when using the profile simulated with 500 mol NO flash$^{-1}$ versus 0 mol NO flash$^{-1}$. In each plot, two of the look-up table inputs are varied and two are held constant. Each plot represents the change averaged over all values of relative azimuth angle (RAA), since RAA has a small impact on the AMF (Fig. S1).

Under both clear and cloudy conditions, the largest differences in AMF between the two profiles are seen at large SZAs

(Fig. 2a and c). This reflects the longer average optical path through the upper troposphere (UT) at larger SZAs, causing greater sensitivity to UT $NO_2$. A similar, though smaller, effect is also seen for larger VZAs.

If viewing geometry is held constant and albedo and surface pressure varied, the largest sensitivity of the AMF to simulated lightning $NO_x$ can be seen at very low albedo and moderate surface pressure ($\sim 860$ hPa) for clear conditions (Fig 2b). The cause for this is illustrated in Fig. 3; Fig. 3c shows how the scattering weight vectors change with albedo and Fig. 3d shows

how they change with surface pressure. Lower albedos yield lower sensitivity to near-surface $NO_2$ (note that scattering weights are proportional to sensitivity) because a photon that reaches the near-surface $NO_2$ will likely be absorbed if it scatters into the surface (Hönninger et al., 2004). The 500 mol flash$^{-1}$ profile does have more $NO_2$ in the boundary layer than the no lightning profile, especially below 900 hPa. This partly balances the increase in UT $NO_2$ from lightning, as there are increases at both low and high sensitivity altitudes. As surface pressure decreases (i.e. higher in elevation), the altitude of minimum sensitivity

moves up. The surface integration limit for Eq. (1) and (3) reduces as well, removing part of the boundary layer profile. Taken





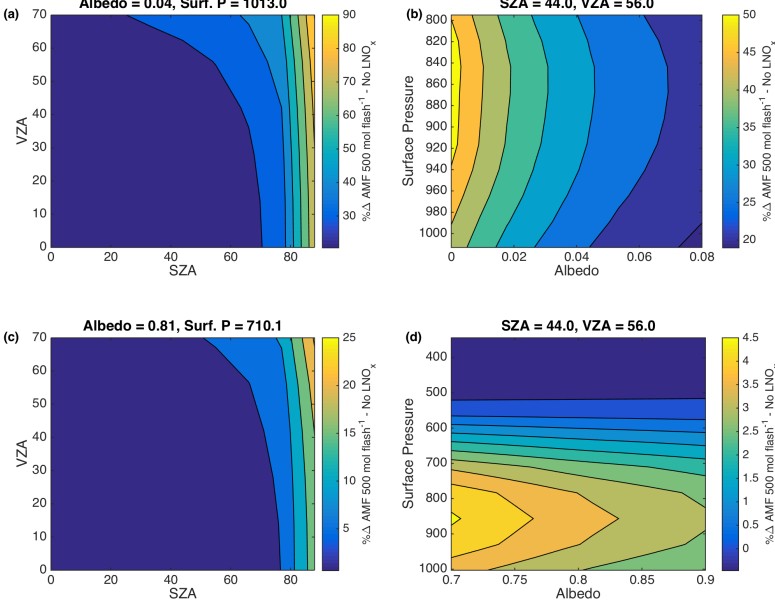

**Figure 2.** Contour plots of the percent change in the AMF when changing from the mean profile without lightning $NO_x$ to the mean with lightning $NO_x$ (500 mol flash$^{-1}$), averaged over the whole WRF-Chem domain. The differences are averaged over all values of RAA. In each plot, two parameters are varied while the other two are held constant. The values of the constant parameters are given above each plot. (a) and (b) use a range of albedos and surface pressure representative of clear pixels; (c) and (d) for cloudy pixels.

together, these changes put more weight on the UT profile and remove the $< 900$ hPa increase that counteracts part of the change in the UT (thus increasing the impact of lightning $NO_x$) until $\sim 860$ hPa. At $\sim 860$ hPa, most of the boundary layer is no longer included in the AMF calculation. Figure 1 shows that above $\sim 800$ hPa, the WRF-Chem profiles start to diverge due to the different amounts of lightning $NO_x$ in each simulation. Therefore, as surface pressure moves above 850–800 hPa, the

5  sensitivity to lightning $NO_x$ begins to decrease because the entire extent of the profile that is integrated changes with changes in the simulated lightning $NO_x$. Since the profile is normalized to the column amount (Eq. 3), only the relative distribution of $NO_2$ matters, and the relative distribution changes very little with the magnitude of lightning $NO_x$ emitted when only considering the part of the profile influenced by lightning $NO_x$.

The effect of changing surface pressure in a regular retrieval will likely be different than that described above, because the

10  above analysis assumes that the profile does not change with surface pressure, where in fact it should, since surface-based emissions will move up with the surface. Consequently, the boundary layer maximum would not be cut off in that case. The effect described here is more consistent with the effect of clouds or an aerosol layer that creates an effectively higher altitude surface (due to scattering), or if using coarse enough a priori profiles that the surface pressure of a pixel is significantly different than the surface pressure in the model used to simulate the profile.



Cloudy conditions exhibit less sensitivity than clear conditions to the amount of lightning $NO_x$ in the modeled profiles due to this shielding effect: in many cases, the cloud is at sufficiently high elevation to obscure the surface $NO_2$ and therefore restricts the profile to the component influenced by lightning $NO_x$. As previously discussed with respect to surface pressure, this means that the relative distribution of $NO_2$ in the visible component of the profile does not change significantly. This is

apparent in Fig. 2, where (c) and (d) show responses roughly $\frac{1}{4}$ and $\frac{1}{10}$ times, respectively, compared to (a) and (b).

Cloudy conditions also tend to have more uniform scattering weights (Fig. 4) due in large part to their high albedo. At high albedo, the probability of "losing" photons to absorption at the surface is significantly reduced, so the reduction in sensitivity towards the surface found with low albedos does not occur. At sufficiently high albedos, there is an enhancement in sensitivity near the surface due to the possibility of extended optical paths near the surface from multiple scattering (Richter and Wagner,

10   2011).

From Fig. 4, it is clear why the impact of lightning $NO_x$ is small in Fig. 2d. For all but the most extreme sun-satellite geometries, the scattering weights are fairly uniform across all altitudes, thus the impact of changes to the relative distribution of $NO_2$ within the UT is minimized since a UV/visible satellite instrument is similarly sensitive to $NO_2$ at any altitude under these conditions. At larger SZAs and VZAs, the cloudy scattering weights do decrease towards the surface because Rayleigh

scattering has a greater effect on the transmitted light along the longer beam paths, scattering photons at higher altitudes and so reducing the fraction of photons observed by the satellite that penetrate to the cloud (Richter and Wagner, 2011). However, the impact is less than in clear conditions. From Fig. 2c, at the largest SZA and VZA simulated, the difference in AMF between the no lightning and 500 mol flash$^{-1}$ profiles is +20–25%—large, but only one-fourth that of clear conditions.

The difference in the AMF obtained using profiles with 665 and 500 mol NO flash$^{-1}$ follows essentially the same pattern

as shown in Figs. 3 and 4, but with $\frac{1}{10}$ to $\frac{1}{5}$ the magnitude (Fig. S2). The only difference in the shape of the contours is that the maximum difference occurs at greater (i.e. lower altitude) surface pressures, because the 665 and 500 mol flash$^{-1}$ profiles are mostly identical in the boundary layer, so the slight countervailing increase in boundary layer $NO_2$ between the 0 and 500 mol flash$^{-1}$ profiles that offset part of the UT increase is not present.

### 3.2   Comparison with observed profiles

Given the large sensitivity of AMFs to the presence of lightning $NO_x$ in the a priori profiles, it is necessary to use a priori profiles that are consistent with observations. Figure 5 compares the average $NO_2$ profile measured in the DC3 campaign (Sect. 2.1) with WRF-Chem profiles averaged along the DC3 flights (Sect. 2.3) for five simulations. It is immediately apparent that the WRF-Chem simulation with no lightning is missing a significant amount of UT $NO_2$ compared to the observed DC3 profile. Both unnudged WRF-Chem simulations with lightning $NO_x$ enabled do qualitatively capture this UT $NO_2$; however

the vertical distribution is biased compared to the DC3 observations with a maximum at 500 hPa not seen in the observed profile and less $NO_2$ between 300–200 hPa than in the observed profile.

We consider how significant these differences between the simulated and observed profiles are in the context of the AMF calculation. To focus only on the effect of the UT profile, we use hybrid profiles. The hybrid profiles for the unnudged 500 mol flash$^{-1}$ are illustrated in Fig. 5b. The free troposphere hybrid uses the DC3 profile up to 750 hPa and the WRF-Chem



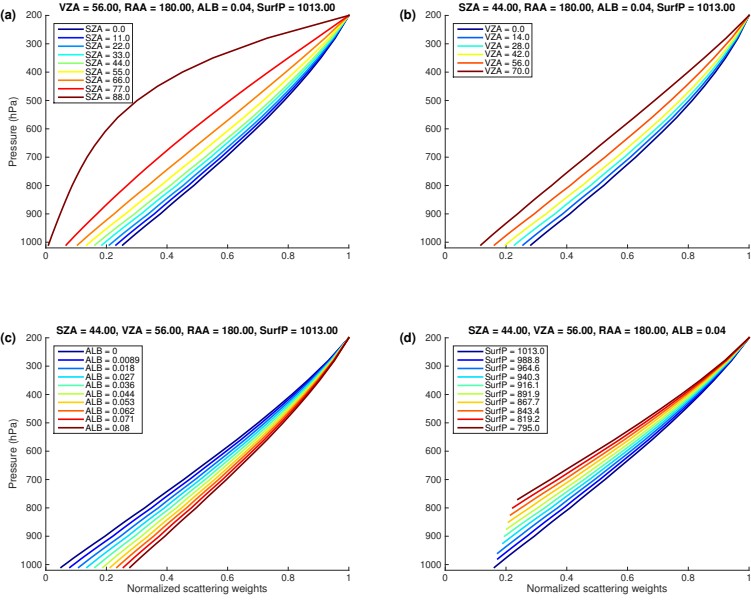

**Figure 3.** Vectors of scattering weights and their variation with each of the four most important look-up table input parameters. Values are representative of clear-sky conditions. Each scattering weight vector is scaled so that the top most entry is 1. Scattering weights are only shown above the surface pressure.

profile above that, while the mid-troposphere hybrid only uses the WRF-Chem profile between 750 and 375 hPa. The free tropospheric hybrid profile focuses on the effect of lightning $NO_2$ on the AMF by removing the difference in the boundary layer between the WRF-Chem and DC3 profiles, while the mid-troposphere hybrid similarly focuses on the effect of the local $NO_2$ maximum around 500 hPa that is not present in the DC3 profile.

5     Table 2 gives the results of AMF sensitivity tests (Sect. 2.5.2) on various hybrid combinations of the profiles in Fig. 5a. We present the average AMF obtained in the sensitivity test using each hybrid profile, and its percent difference relative to the mean AMF obtained using the DC3 profile. Because OMI experiences a more limited range of solar zenith angles during summer over the US ($\sim 30° \pm 6°$, on average) than are defined in the TOMRAD look-up table, we also compare a subset of the AMF sensitivity tests with the SZA $< 40°$.

10     Comparing the 0 mol flash$^{-1}$ WRF-Chem profiles to the DC3 profile, we see that difference $NO_2$ above 375 hPa has a large impact on the AMF, causing a 26–35% low bias in the AMF, depending on the SZAs considered. Adding lightning $NO_x$ to the WRF-Chem simulation (the 500 and 665 mol flash$^{-1}$ profiles) corrects this bias. Recent work (Nault et al., submitted) suggests that the previous mean value of mol NO flash$^{-1}$ (500 mol flash$^{-1}$) is 33% low; comparing the AMFs obtained from profiles generated with 500 and 665 mol flash$^{-1}$ changes the sign of the AMF bias relative to the DC3 profile, but not its magnitude.





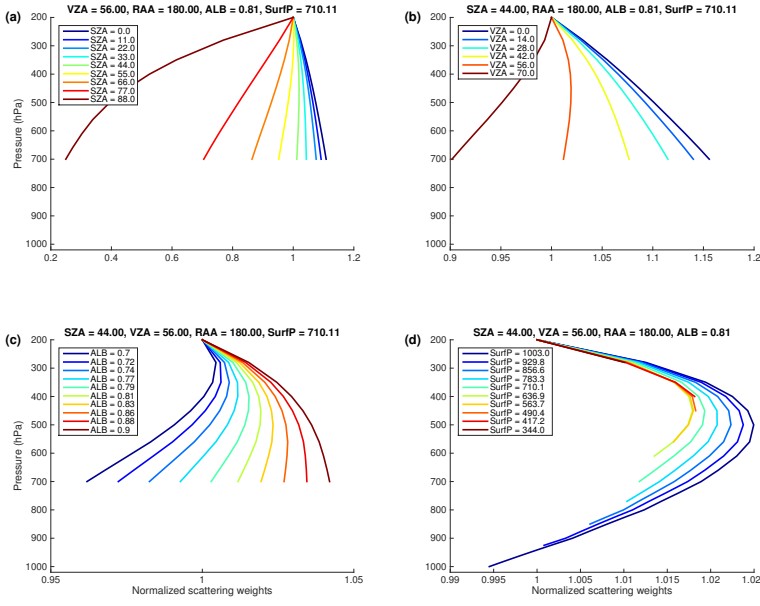

**Figure 4.** As in Fig. 3, but for cloudy conditions. Note that the $x$-axis limits are different from Fig. 3 and each other.

The purpose of including the mid-troposphere hybrid profiles, which only use the WRF-Chem profile between 700 and 375 hPa, is to evaluate the impact of the simulated $NO_2$ maximum around 500 hPa. In almost all cases, the bias of these hybrid profiles against the DC3 profile is less than the corresponding free-troposphere hybrid. Thus, that anomalous maximum at 500 hPa has a smaller impact than the overall presence or absence of lightning $NO_2$, as one would expect.

An additional complication arises when considering the effect of nudging the model meteorology. By default, the meteorology in WRF is driven by the model's internal physics and is constrained by reanalysis meteorology only through the initial and boundary conditions. WRF has the option, however, to constrain meteorology throughout the domain using four dimensional data analysis (FDDA) nudging. Temperature and water vapor mixing ratio can both be nudged, and both are used in the Grell 3D cumulus physics calculation in WRF (Grell, 1993; Grell and Dévényi, 2002), which outputs the cloud top height that is

used by the Price and Rind (1992) parameterization of flash rate.

With FDDA nudging, lightning flash rates throughout the domain decreased by approximately a factor of 2 compared to the unnudged case (Fig. S3). Comparing both temperature and water vapor mixing ratios from nudged and unnudged simulations, we find that nudged and unnudged temperature profiles only differ by $\sim$ 1–2 K at each model level on average, and both agree well with DC3 measurements. The water vapor profiles change more significantly, and the profiles resulting from the nudged

simulation agree better with those measured during DC3 (Fig. S4). Therefore, we conclude that the changes to the water vapor profiles are responsible for the 2x change in lightning flash rates.



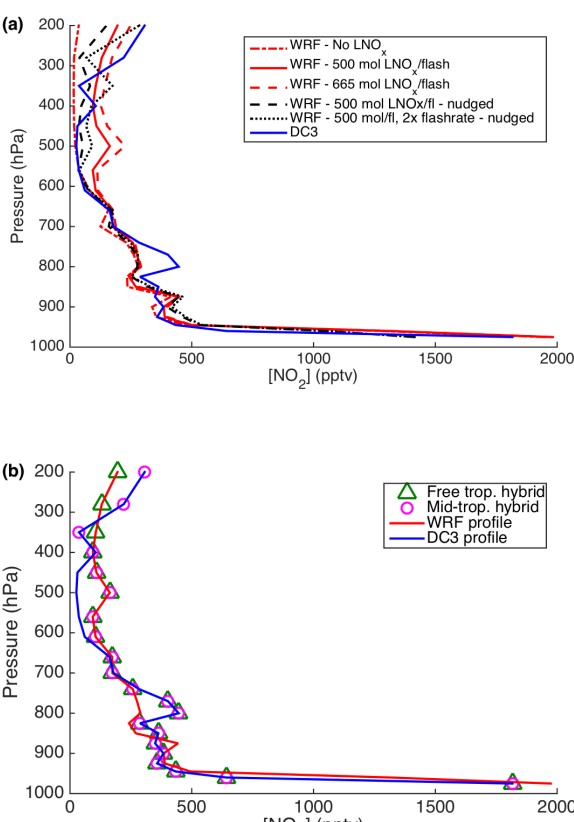

**Figure 5.** (a) Comparison of the $NO_2$ profiles obtained from binning all DC3 data and WRF-Chem output along the DC3 flight track (Sect 2.3) to pressure bins centered on the pressure the scattering weights are defined at. (b) The binned DC3 and WRF-Chem (500 mol flash$^{-1}$, no nudging) profiles; green triangles mark pressure levels from each profiles used in the free troposphere hybrid profile, magenta circles mark pressure levels used in the mid-troposphere hybrid profile.





| Profile | Avg. AMF | %$\Delta$ AMF vs. DC3 | Avg. AMF SZA $< 40°$ | %$\Delta$ AMF(SZA $< 40°$) vs. DC3 |
|---|---|---|---|---|
| DC3 | 1.56 | – | 1.30 | – |
| Free Trop. Hybrid-0 | 1.02 | −34.87 | 0.96 | −26.01 |
| Mid. Trop. Hybrid-0 | 1.55 | −0.87 | 1.29 | −1.10 |
| Free Trop. Hybrid-500 | 1.51 | −3.40 | 1.29 | −0.89 |
| Mid. Trop. Hybrid-500 | 1.60 | 2.26 | 1.34 | 2.86 |
| Free Trop. Hybrid-665 | 1.62 | 3.54 | 1.36 | 4.70 |
| Mid. Trop. Hybrid-665 | 1.62 | 3.44 | 1.36 | 4.24 |
| Free Trop. Hybrid-500, nudge | 1.26 | −19.37 | 1.12 | −13.93 |
| Mid. Trop. Hybrid-500, nudge | 1.56 | −0.08 | 1.30 | −0.08 |
| Free Trop. Hybrid-500, nudge, 2x flashrate | 1.48 | −5.45 | 1.26 | −3.05 |
| Mid. Trop. Hybrid-500, nudge, 2x flashrate | 1.58 | 1.18 | 1.32 | 1.33 |

**Table 2.** Results of the AMF sensitivity tests on the hybrid profiles in Fig. 5

Using the $NO_2$ profiles resulting from the nudged simulation with 500 mol flash$^{-1}$, we see in Fig. 5a that there is significantly less simulated $NO_2$ near 200 hPa than in the unnudged run and the DC3 observations. The AMF sensitivity tests show that this reintroduces a 14–20% low bias compared to the AMF derived from the DC3 profile, a significant increase in the bias compared to the unnudged simulation. Doubling the flash rate largely corrects this bias by increasing the $NO_2$ found in the upper part of the profile (Fig. 5a).

### 3.3 Effect of varied lightning emissions on BEHR AMFs

To illustrate the impact of missing lightning $NO_x$ on a full retrieval, we use the unnudged WRF-Chem $NO_2$ profiles simulated with 0, 500 and 665 mol NO flash$^{-1}$ as a priori profiles in the BEHR retrieval and examine the change in both AMF and retrieval $NO_2$ vertical column density (VCD) with the change in simulated lightning $NO_x$.

Figure 6 shows the average percent change in AMFs (top) and absolute change in VCDs (bottom) between retrievals using profiles generated using 0 and 500 mol NO flash$^{-1}$ (Fig. 6a, c) and between 500 and 665 mol NO flash$^{-1}$ (Fig. 6b, d). These results were obtained by averaging data from 18 May to 23 June 2012, treating the data as described in Sect. 2.5.1.

Most importantly, we see in Fig. 6a that the change due to the inclusion of lightning $NO_2$ is not constant throughout the domain, but is regionally specific. The SE US sees the greatest change in AMF, as it has very active lightning (Hudman et al., 2007). This leads to changes in the retrieved VCD of 1 to $2 \times 10^{15}$ molec. cm$^{-2}$.

We consider two uncertainty values to determine if this change is significant. Bucsela et al. (2013) calculated a global mean uncertainty of $1 \times 10^{15}$ molec. cm$^{-2}$ for tropospheric $NO_2$ VCDs. Boersma et al. (2004) calculated a typical uncertainty of 23% in tropospheric AMFs for polluted conditions. Since, on average, $32 \pm 6$ (mean $\pm 1\,\sigma$) pixels contribute to each value in our average, the reduced uncertainty is $\sim 0.2 \times 10^{15}$ molec. cm$^{-2}$ and 4%, respectively. The changes we find in the tropospheric





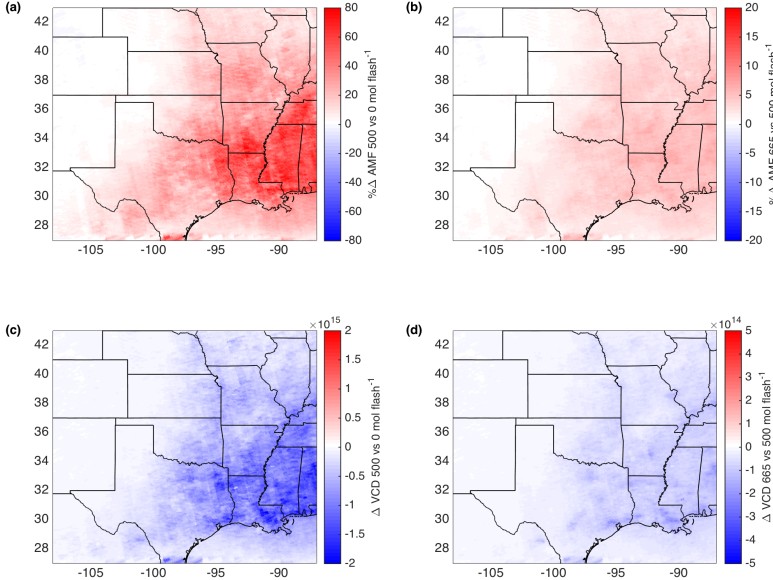

**Figure 6.** Average percent difference in AMFs (a,b) and absolute difference in VCDs (c,d) averaged over the time period 18 May–23 June 2012. (a,c) Difference between profiles generated using 500 mol NO flash$^{-1}$ and 0 mol NO flash$^{-1}$; (b,d) Difference between profiles generated using 665 mol NO flash$^{-1}$ and 500 mol NO flash$^{-1}$. Note that in (c) and (d) the color scale is one-fourth that of (a) and (b).

VCD due to the inclusion or exclusion of lightning $NO_2$ from the a priori profiles exceed that uncertainty in $\sim 50\%$ of the domain; the changes in the AMF exceed the uncertainty in $\sim 70\%$ of the domain.

The effect on the retrieval from increasing the mol NO flash$^{-1}$ from 500 to 665 is about 5–10x smaller, as seen in Fig. 6b, d. In Fig. 1 and Fig. 5, we saw that the change in the UT profile was smaller when increasing the mol flash$^{-1}$ from 500 to 665 compared to increasing from 0 to 500 as expected. The nonlinear nature of the AMF calculation also contributes to the smaller change in AMFs and VCDs between 500 and 665 mol flash$^{-1}$ profiles; as the contribution of lightning $NO_2$ increases, both the numerator (at the relevant pressure levels) and denominator of Eq. (3) increase. The increasing denominator will cause the same magnitude increase in the numerator to have a smaller effect on the overall AMF.

## 4  Discussion

Accurately representing lightning $NO_2$ in a priori profiles for retrieval of $NO_2$ from space is vital not only when retrieving lightning events, but any retrieval in a region and time period influenced by lightning. While work from the DC3 campaign has shown that the lifetime of $NO_x$ in the near field of thunderstorms is remarkably short ($\sim 3$ h, Nault et al., 2016) due to active chemistry with peroxy radical species convected from the surface, once those peroxy radicals are depleted, the UT lifetime



of $NO_x$ in the far-field from thunderstorms is in the range of 0.5 to 1.5 days (Bertram et al., 2007; Fried et al., 2008; Apel et al., 2012; Nault et al., 2016). As shown in Sect. 3.1, this means that the presence or absence of lightning $NO_2$ in the a priori profiles has a large effect on the retrieval AMFs in clear-sky conditions which are used to obtain information about boundary layer $NO_x$ (e.g. Lamsal et al., 2010; Beirle et al., 2011; Valin et al., 2013; Lamsal et al., 2015; Lu et al., 2015; Liu et al., 2016,
5    2017).

We note that the WRF-Chem model used here may not be adequately capturing this near-field chemistry as the simulated concentrations of methyl peroxy nitrate (MPN) are significantly lower than those measured by the DC3 campaign, particularly in the range of 300 to 400 hPa. We suspect that modeled concentrations of the methyl peroxy radical precursor are too low, but have not investigated this. However, we do not believe this significantly impacts our conclusions, as when we bin the DC3
MPN data as in Fig. 5, the MPN concentration is $\frac{1}{5}$ to $\frac{1}{10}$ that of $NO_x$, so the effect on the AMF is expected to be less than the effect of increasing the modeled mol NO flash$^{-1}$ from 500 to 665.

### 4.1 Effect of nudged meteorology on flash counts

Additionally, the results from Sect. 3.2 regarding the reduction in number of lightning flashes when using FDDA nudging towards the NARR reanalysis are particularly relevant, as Laughner et al. (2016) showed the importance of using daily, high-
spatial resolution a priori profiles to accurately resolve differences in $NO_2$ VCDs upwind and downwind of a city, and suggested the use of nudging to reduce the uncertainty due to wind direction, especially. Our results here indicate that (1) missing lightning $NO_2$ in the a priori profiles will lead to large overestimations of VCDs, which, among other things, would lead to overestimates of $NO_x$ emissions based on such a retrieval, and (2) that when using nudging within a WRF-Chem simulation to constrain the meteorology, its effect on lightning flash rates must be checked to ensure it does not inadvertently affect the upper tropospheric
$NO_2$ profile.

Nevertheless, although our results showed that the $NO_2$ profile resulting from the nudged run without doubled flash counts had less UT $NO_2$ that the average DC3 profile, we cannot conclude that the flash rates calculated with nudged meteorology are underestimated, particularly as Wong et al. (2013) found the opposite result when comparing to the National Lightning Detection Network. A direct comparison with Wong et al. (2013) is complicated by the different choices of model options
(such as cumulus physics: Grell 3D in ours vs. Grell-Devenyi in Wong; Lin vs. Thompson microphysics; NARR vs. NCEP Global Forecasting System Final meteorology). A full analysis of the reason that activating FDDA nudging causes the flash rates to decrease by 50% in our case is beyond the scope of this paper.

### 4.2 Relevance to cloud slicing

In the context of work using cloud-slicing techniques to derive $NO_2$ profiles (e.g. Choi et al., 2014), our results suggest that
profile shape is a minor contribution to the uncertainty. By using a simulated retrieval with a known $NO_2$ concentration profile, Choi et al. (2014) estimated 20–30% uncertainty in the $NO_2$ concentration derived from their cloud-slicing approach. Our work here shows that, for fully cloudy conditions, the change in the AMF between a no lightning and 500 mol flash$^{-1}$ $NO_2$ profile is $\leqslant 5\%$ (Sect. 3.1); since Choi et al. (2014) used a typical C-shaped $NO_2$ profile that included lightning $NO_2$ (e.g.





Pickering et al., 1998), any uncertainty should be closer to the difference we observed between the 500 and 665 mol flash$^{-1}$ profiles, $\leqslant 1\%$.

### 4.3 Relevance to global and geostationary retrievals

To the best of our knowledge, the chemical transport models used to generate the a priori profiles in the NASA Standard Product

and KNMI DOMINO product for OMI $NO_2$ include lightning $NO_x$ in the simulation. However, for researchers wishing to generate high spatial resolution a priori profiles using models such as WRF-Chem or the Community Multiscale Air Quality (CMAQ) model that have thus far focused on lower troposphere chemistry for air quality implications, it is important to verify whether that model setup includes lightning $NO_x$. Retrievals that use a priori profiles without a lightning $NO_x$ parameterization will suffer from a regionally dependent, systematic positive bias in retrieved VCDs. This is particularly difficult to account for

given that the bias is unlikely to be reduced by averaging, nor is it constant enough spatially to be addressed as a coarse, ad hoc correction to the AMF.

The next generation of polar orbiting (TROPOMI) and geostationary (TEMPO, Sentinel-5, GEMS) UV-visible spetrometers will have even greater spatial resolution than OMI. To get the most value out of these high spatial resolution detectors, high spatial and temporal resolution a priori profiles are necessary (e.g. Russell et al., 2011; Laughner et al., 2016; Goldberg et al.,

2017). High resolution air quality models, such as WRF-Chem or CMAQ, are one avenue to produce a priori profiles with resolution of 1 to 10 km. Ensuring that lightning $NO_x$ is adequately parameterized in the models is essential for any retrieval, but especially geostationary satellites such as TEMPO, which will retrieve $NO_2$ at larger solar zenith angles than polar orbiting satellites. At such large SZAs, the relative importance of accurate UT $NO_2$ profiles is even greater than for OMI retrievals.

## 5 Conclusions

We quantify the impact of lightning $NO_2$ on a priori profiles used in satellite retrievals of $NO_2$. We find that, on average, compared to an average $NO_2$ profile constructed from measurements taken during the DC3 campaign, excluding lightning $NO_2$ leads to a $-35\%$ bias in the AMF if all solar zenith angles are considered, and $-26\%$ for solar zenith angles relevant to the OMI instrument in the summer. We find that, using the Price and Rind (1992) parameterization in WRF-Chem with the Grell-3D cumulus model, 500 to 665 mol NO flash$^{-1}$ yields AMFs within 5% of those obtained using the DC3 profile. We

also find that, if FDDA nudging is used, flash rates must be multiplied by a factor of 2 to get the same agreement with this model configuration.

Implementing profiles generated with 0, 500, and 665 mol NO flash$^{-1}$ in the BEHR retrieval, we find that the effect on the AMF is very regionally dependent. Changing from profiles using 0 mol NO flash$^{-1}$ to 500 mol NO flash$^{-1}$ shows the largest increase in the AMF of $\sim 80\%$ occurring in the SE US. This results in changes to the VCD of 1 to $2 \times 10^{15}$ molec. cm$^{-2}$. The

effect is nearly 0 on the west edge of the domain, over the Rocky Mountains. Further increasing the mol NO flash$^{-1}$ from 500 to 665 only results in a $\sim 5\%$ change to the AMF.





*Code and data availability.* The AutoWRFChem code used to automate the preparation of meteorological and chemical inputs and execution of WRF-Chem is available at https://github.com/CohenBerkeleyLab/AutoWRFChem-Base (Laughner, 2017b). The versions of WRF-Chem v3.5.1, WPS v3.5.1, NEI conversion utility, MEGAN biogenic model, and MOZBC utility with the modification to handle the R2SMH chemical mechanism and corresponding emissions are available at https://github.com/CohenBerkeleyLab/AutoWRFChem-R2SMH, v1.0.0.

The retrievals used in Section 3.3 are available at https://doi.org/10.6078/D19S9D (Laughner, 2017c). The analysis code and WRF-Chem namelist files are available at https://doi.org/10.5281/zenodo.834746 (Laughner, 2017a). For access to the BEHR algorithm contact the corresponding author, R.C. Cohen.

*Competing interests.* The authors declare no competing interests.

*Acknowledgements.* The authors gratefully acknowledge support from the NASA ESS Fellowship NNX14AK89H, NASA grant NNX15AE37G,

and the TEMPO project grant SV3-83019. The MODIS Aqua L2 Clouds 5-Min Swath 1 and 5km (MYD06_L2) and MODIS Terra+Aqua Albedo 16-Day L3 Global 0.05Deg CMG V005 were acquired from the Level-1 and Atmospheric Archive and Distribution System (LAADS) Distributed Active Archive Center (DAAC), located in the Goddard Space Flight Center in Greenbelt, Maryland (https://ladsweb.nascom.nasa.gov/). We acknowledge use of the WRF-Chem preprocessor tools mozbc, fire_emiss, etc. provided by the Atmospheric Chemistry Observations and Modeling Lab (ACOM) of NCAR. This research used the Savio computational cluster resource provided by the Berkeley Research

Computing program at the University of California, Berkeley (supported by the UC Berkeley Chancellor, Vice Chancellor of Research, and Office of the CIO). The authors also wish to thank Mary Barth for assistance with the lightning $NO_x$ module in WRF-Chem.



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
