# Peer review of "Quantification of the effect of modeled lightning NO2 on UV-visible air mass factors"

_Atmospheric Measurement Techniques, 2017_

## Referee Comment (RC1) · Anonymous Referee #1 · 13 Sep 2017

In this manuscript, Laughner and Cohen quantify and discuss the impact of lightning NO2 on a priori profiles used in satellite retrievals of tropospheric NO2 columns. The authors note that high resolution air quality models such as CMAQ and WRF-Chem may be preferred in some retrieval algorithms due to their spatial resolution, but these models may not represent lightning sources accurately (if at all). They find that ignoring lightning NO2 can result in substantial underestimates in the calculated air mass factors, mostly in cases with large viewing geometries (including those that will be encountered in forthcoming geostationary observations) due to the relatively longer optical path through the upper troposphere. The effect of accurately including lightning NO2 is most important under clear conditions given the strong function of scattering

weights with altitude, while the error becomes smaller for cloudy pixels due to fairly uniform above-cloud scattering weights. The biases that result from ignoring lightning NO2 are regionally dependent, cannot be reduced by averaging, and are sufficiently heterogeneous in space that even a coarse ad-hoc correction to the AMF cannot account for the issue.

Overall, this manuscript is well-written and presents a cogent argument. It is extremely detailed but exceedingly clear, and should be of interest to the satellite remote sensing community. In my opinion, publication in AMT is certainly warranted, and I have only very minor suggestions.

1) I think adding a reference to Travis et al., ACP, 2016 (https://www.atmos-chem-phys.net/16/13561/2016/) somewhere in the introduction is important. That paper also called specific attention to the importance of upper troposphere NO2 from lightning, noting that, if ignored, it can lead to underestimates of the air mass factor particularly over the Southeast US. This seems relevant to the manuscript here, and the present authors are listed as co-authors.

2) In Section 2.2, could the authors include a statement about the vertical resolution with which WRF was run?

3) In Section 2.5.1, the authors note they have used slant column densities from the NASA Standard Product version 2 (SP v2). Krotkov et al. (2017) recently published documentation for the latest NASA release (SP v3), which includes a new spectral fitting algorithm, and higher resolution profiles. Of course, I don't anticipate the major conclusions of this paper to change. But could the authors include a short statement/argument to reassure the audience this is expected to be the case?

4) Also in Section 2.5.1, the authors note the use of TOMRAD look-up tables for the pressure-dependent scattering weights. Can anyone access this look-up table data, or is this provided by request from Bucsela et al. specifically for the BEHR retrieval? If the former, could the authors include the location of where this open data could be

accessed by others for their own work?

5) I am curious whether the default BEHR NO2 retrieval, which uses WRF-Chem model profiles, currently accounts for lightning. I understand this is not directly relevant to the present study (since new a-priori profiles are tested in this case), but it might be useful to point out to the readers, given the authors acknowledge that WRF-Chem is often run by default without lightning NO2.

---

## Referee Comment (RC2) · Anonymous Referee #2 · 19 Sep 2017

The authors present a sensitivity study examining the impact of lightning NOx in the BEHR product. As the authors discuss, this is likely not a major concern for operational NO2 products as most operational algorithms rely on a priori profiles taken from global models, which generally include various emission sources including lightning NOx. Simulations from regional air quality models may not include certain emission sources such as lightning NOx as they have little impact on surface level concentrations. Lately, there have been some efforts to use high-resolution a priori information in satellite (e.g., OMI) NO2 retrievals, recognizing profile sensitivity of retrievals and use of course resolution model profiles by operational algorithms. This paper is providing an example that using high-resolution a priori profiles does not necessarily improve

[Figure]

NO2 retrievals, but rather it could add considerable errors in the product. I would be surprised if this is any new piece of information for those involved in operational algorithm development.

This reviewer is aware of the BEHR product and has used it for own research, partly due to the team's assertion that the BEHR NO2 product is highly precise and accurate. In their first paper (Russell et al., 2011), the claim was that the impact of lightning NOx is minimal and could be neglected. In fact, I just discovered that one of the reviewers of the Russell et al., (2011) paper expressed a major concern about the neglect of lightning NOx in their simulation which apparently was dismissed by suggesting that the effect is negligible. As a user of the product, this paper from the same group is very frustrating. If this is still the case in their product, it should be discussed in this paper so that the data users are aware of the error/deficiency in the BEHR product.

I feel that this is a lengthy paper based on very limited research. Large part of the discussion on scattering weights may be a self-educating piece for the authors. The discussion section (Section 4) is either inconclusive or conclusions drawn based on limited research. Due to several major concerns I do not think this paper, in the current form, would add any significant insights to the AMT readership.

Major concerns:

1) Lightning NOx emissions vary strongly both spatially and seasonally. Therefore, retrievals are affected differently for various seasons. Conclusions drawn from ∼1 month in late-spring/early-summer will most likely be incomplete and misleading. I strongly recommend expanding this analysis over the continental US (analyzing urban vs. rural, east vs. west, north vs. south, etc.) for all months. It would be more instructive if this could be discussed in global context.

2) Model vs measurements discrepancies for the upper tropospheric NO2 are bundled solely to lightning NOx emissions and therefore are attempted to address by using a fixed NO mol/flash. There are wide varieties of estimates in the literature, going as low

as a factor of 5 lower than the estimate used in this paper. I think, the authors should perform similar analysis with additional simulations for confidence in the presented results.

3) I am also concerned by the incomprehensive nature of this analysis. The BEHR algorithm is based on several inputs from the operational OMNO2 products, for instance the use of stratospheric NO2 estimates from OMNO2 for calculating tropospheric slant column amount. Modulation of upper tropospheric NO2 should lead to different estimates for stratospheric NO2. Could you do your own stratosphere-troposphere separation and quantify the effect on both stratospheric and tropospheric NO2 estimates?

Minor comments:

4) Page 2, lines 27-33: Wouldn't the lifetime of NOx vary with altitude? Suggesting the upper tropospheric NOx lifetime < 4 days might be misleading.

5) Page 4, line 17, Section 2.3: What is the altitude range of DC3 measurements? Discuss how you treat simulated profile beyond the range of DC3 measurements.

6) Page 5, line 19: How was the estimate of ghost column made? Does this mean, you add a-priori-derived NO2 columns below the cloud to the retrieved tropospheric columns? How does this approach compare with the operational (DOMINO, OMNO2) procedures?

7) Page 5, line 24: What is the logic behind using the fixed tropopause pressure at 200 hPa? Does that mean the BEHR NO2 columns represent columns below 200 hPa? How do you deal with possible errors from using OMNO2-based stratospheric NO2 columns that is likely based on (variable) meteorology-based tropopause pressures?

8) Page 6, Eqn 4: Here and everywhere else in the text. This should be tropospheric AMF, not total AMF. Correct or clarify this.

9) Page 6, line 16: Use of black-sky albedo instead of Lambert-Equivalent Reflectivity (LER) should be a large source of errors. Please comment on this based on some

recent publications (e.g. Lin et al., 2015; Vasilkov et al., 2017).

10) Page 6, line 18-20: Will this approach capture the seasonal variation of surface pressure? How big is its effect on AMF?

11) Page 6, line 22: "OMI cloud fraction...". Is this "effective" or "radiative" cloud fraction?

12) Page 6-7, Section 2.5.2: Related to "Surface pressure (cloudy)". I cannot understand the logic of having different surface pressures for clear and cloudy pixels? Should not this be cloud pressure instead?

13) Page 8, Figure 1: Please, also include profile shapes which might be more relevant for AMF.

14) Page 9, lines: 1:3: This discussion is confusing. Should not the effect be based on the altitude of lightning generated NOx?

15) Page 10, line 2: In "obscure the surface NO2...", don't you mean "below cloud"?

16) Page 16, Section 4.1: What is the message of this section? Why mention nudging at all in the paper if understanding of the 50% decrease in the flash rates by activating FDDA nudging is beyond the scope of the paper?

17) Page 16, line 22: In "less UT NO2 that" => "less UT NO2 than"

18) Page 16, Section 4.2: To estimate uncertainty in cloud slicing, Choi et al. 2014 might have conducted more comprehensive analysis, considering errors in cloud and other parameters.

---

## Author Comment (AC1) · 4 Oct 2017

Please see the attached PDF for our response to both reviewer's comments.

Please also note the supplement to this comment:
https://www.atmos-meas-tech-discuss.net/amt-2017-263/amt-2017-263-AC1-supplement.pdf

———————————————————

---

## Author Response (AR1)

**Quantification of the effect of modeled lightning NO$_2$ on UV-visible air mass factors**

**Response to Anonymous Referee #1**

Joshua L. Laughner and Ronald C. Cohen

October 4, 2017

We thank the reviewer for their positive comments. The individual corrections suggested are addressed below. The reviewer's comments will be shown in red, our response in blue, and changes made to the paper are shown in black block quotes. Unless otherwise indicated, page and line numbers correspond to the original paper. Figures, tables, or equations referenced as "R$n$" are numbered within this response; if these are used in the changes to the paper, they will be replaced with the proper number in the final paper. Figures, tables, and equations numbered normally refer to the numbers in the original discussion paper.

1) I think adding a reference to Travis et al., ACP, 2016 (https://www.atmos-chem-phys.net/16/13561/2016/) somewhere in the introduction is important. That paper also called specific attention to the importance of upper troposphere NO2 from lightning, noting that, if ignored, it can lead to underestimates of the air mass factor particularly over the Southeast US. This seems relevant to the manuscript here, and the present authors are listed as co-authors.

We have added a reference to Travis et al. (2016) on p. 2, l. 26.

2) In Section 2.2, could the authors include a statement about the vertical resolution with which WRF was run?

We have added:

> "across a domain that covers the same region as the DC3 campaign at 12 km model resolution **with 29 vertical levels**."

3) In Section 2.5.1, the authors note they have used slant column densities from the NASA Standard Product version 2 (SP v2). Krotkov et al. (2017) recently published documentation for the latest NASA release (SP v3), which includes a new spectral fitting algorithm, and higher resolution profiles. Of course, I dont anticipate the major conclusions of this paper to change. But could the authors include a short statement/argument to reassure the audience this is expected to be the case?

We have added the following text to the second paragraph in Sect. 2.5.1:

> "Version 3 of the NASA Standard Product was released in 2016, and includes new spectral fitting and tropospheric AMF calculations. The change from SP v2

to v3 does not affect any of the AMF calculations in this work. Krotkov et al. (2017) indicates that the tropospheric VCDs over unpolluted areas are similar between SP v2 and v3, therefore, when effects on retrieved VCDs are considered here, we expect our conclusions to be unaltered when BEHR is updated to use SP v3 data."

4) Also in Section 2.5.1, the authors note the use of TOMRAD look-up tables for the pressure-dependent scattering weights. Can anyone access this look-up table data, or is this provided by request from Bucsela et al. specifically for the BEHR retrieval? If the former, could the authors include the location of where this open data could be accessed by others for their own work?

We have added the LUT to the analysis code respository, and mentioned this in the Code and Data Availability section.

5) I am curious whether the default BEHR NO2 retrieval, which uses WRF-Chem model profiles, currently accounts for lightning. I understand this is not directly relevant to the present study (since new a-priori profiles are tested in this case), but it might be useful to point out to the readers, given the authors acknowledge that WRF-Chem is often run by default without lightning NO2.

The current BEHR profiles do not include lightning $NO_2$, which was part of the motivation for this study. We have added the following sentence to the first section of the discussion:

> "These new a priori profiles will correct the absence of modeled lightning $NO_2$ in the v2.1C a priori profiles."

**References**

Krotkov, N. A., Lamsal, L. N., Celarier, E. A., Swartz, W. H., Marchenko, S. V., Bucsela, E. J., Chan, K. L., Wenig, M., and Zara, M.: The version 3 OMI $NO_2$ standard product, Atmos. Meas. Tech., 10, 3133–3149, doi:10.5194/amt-10-3133-2017, URL `https://www.atmos-meas-tech.net/10/3133/2017/`, 2017.

Travis, K. R., Jacob, D. J., Fisher, J. A., Kim, P. S., Marais, E. A., Zhu, L., Yu, K., Miller, C. C., Yantosca, R. M., Sulprizio, M. P., Thompson, A. M., Wennberg, P. O., Crounse, J. D., St. Clair, J. M., Cohen, R. C., Laughner, J. L., Dibb, J. E., Hall, S. R., Ullmann, K., Wolfe, G. M., Pollack, I. B., Peischl, J., Neuman, J. A., and Zhou, X.: Why do models overestimate surface ozone in the Southeast United States?, Atmos. Chem. Phys., 16, 13 561–13 577, doi:10.5194/acp-16-13561-2016, URL `https://www.atmos-chem-phys.net/16/13561/2016/`, 2016.

**Quantification of the effect of modeled lightning NO$_2$ on UV-visible air mass factors**

**Response to Anonymous Referee #2**

Joshua L. Laughner and Ronald C. Cohen

October 4, 2017

We thank the reviewer for their detailed reading of the paper. Below, we address each concern individually. The reviewer's comments will be shown in red, our response in blue, and changes made to the paper are shown in black block quotes. Unless otherwise indicated, page and line numbers correspond to the original paper. Figures, tables, or equations referenced as "R$n$" are numbered within this response; if these are used in the changes to the paper, they will be replaced with the proper number in the final paper. Figures, tables, and equations numbered normally refer to the numbers in the original discussion paper.

I feel that this is a lengthy paper based on very limited research. Large part of the discussion on scattering weights may be a self-educating piece for the authors. The discussion section (Section 4) is either inconclusive or conclusions drawn based on limited research. Due to several major concerns I do not think this paper, in the current form, would add any significant insights to the AMT readership.

We respectfully disagree with the reviewer's conclusion that this paper does not add significant insights to the AMT readership. While the conclusion that the presence or absence of lightning NO$_2$ in the a priori profiles significantly affects the tropospheric AMFs may not be a surprising result, we know of no other work that has explicitly demonstrated this effect, especially in retrievals using high spatial resolution chemical transport models. Goldberg et al. (2017) considered lightning as a possible cause for the difference between GMI and CMAQ UT NO$_2$ profiles, but did not explicitly demonstrate the effect of varying lightning NO$_x$ emissions in the CTM.

We believe that the detailed discussion of the scattering weights helps the reader understand the changes to the AMFs under different conditions shown in Fig. 2.

For the discussion, we recognize that ending the first two subsections with caveats may take away from the stronger conclusion presented in the first paragraph of each subsection. We have reordered these two subsections to place the stronger conclusion at the end of each section.

1) Lightning NOx emissions vary strongly both spatially and seasonally. Therefore, retrievals are affected differently for various seasons. Conclusions drawn from $\sim$ 1 month in late-spring/early-summer will most likely be incomplete and misleading. I strongly recommend expanding this analysis over the continental US (analyzing urban vs. rural, east vs.

west, north vs. south, etc.) for all months. It would be more instructive if this could be discussed in global context.

We appreciate the reviewer's interest in understanding the effects on lightning $NO_2$ in broader regions and time periods. Our goal at this time was to first verify that, at a time and place where lightning activity is expected, that the impact on high-resolution a priori profiles is significant and to identify a model configuration that produces a good representation of lightning $NO_2$ in the a priori profiles. For that reason, we chose to focus on a spatial domain and time period where in situ observations were available in the form of the DC3 campaign. We intend to apply the lessons learned here to the next generation of the BEHR retrieval.

To avoid giving the impression that quantitative results apply without exception to other time periods, we have made the following changes. In the abstract:

> "...Focusing on **late spring and early summer in** the central and eastern United States, we find that a simulation without lightning..."

In the first section of the discussion:

> "...the presence or absence of lightning $NO_2$ in the a priori profiles has a large effect on the retrieval AMFs in clear-sky conditions which are used to obtain information about boundary layer $NO_x$ (e.g. Lamsal et al., 2010; Beirle et al., 2011; Valin et al., 2013; Lamsal et al., 2015; Lu et al., 2015; Liu et al., 2016, 2017). **Since many of these studies focus on summer months when thunderstorms are common over the US (Barth et al., 2015), the inclusion of lightning $NO_2$ in the a priori profiles is necessary to accurately constrain the emissions. Lightning is less frequent in wintertime, but the southeast US does experience winter lightning (Orville et al., 2001; Hunter et al., 2001). Therefore, wintertime retrievals will likely see significantly less but nonzero impact from the inclusion of lightning $NO_2$ in the a priori profiles. Future work will verify this as new a priori profiles are planned for inclusion in the next generation of the BEHR retrieval.**"

In the conclusion:

> "...we find that the effect on the AMF is very regionally dependent. **For summertime retrievals, changing** from profiles using 0 mol NO flash$^{-1}$ to 500 mol NO flash$^{-1}$ shows the largest increase in the AMF..."

Regarding global context, recent work (e.g. Cooper et al., 2014; Nault et al., 2017) suggests that regionally-specific lightning parameterizations are necessary to accurately capture the behavior of lightning around the world. We hope that our work helps other groups developing high resolution retrievals understand the magnitude of the sensitivity of $NO_2$ retrievals to lightning $NO_2$ and presents an approach to quantify the uncertainty due to the model's lightning parameterization as in situ observations to compare against become available in other regions.

2) Model vs measurements discrepancies for the upper tropospheric NO2 are bundled solely to lightning NOx emissions and therefore are attempted to address by using a fixed NO mol/flash. There are wide varieties of estimates in the literature, going as low as a factor of 5 lower than the estimate used in this paper. I think, the authors should perform similar analysis with additional simulations for confidence in the presented results.

From Table 2, we can see that the AMFs calculated using the free tropospheric hybrid profiles from the 500 and 665 unnudged mol flash$^{-1}$ simulations bracket the AMF calculated from the average DC3 profile, while the simulation with 0 mol flash$^{-1}$ yields an AMF 35% low. Since the increase in AMF with lightning emissions is approximately linear, this suggests that lower mol flash$^{-1}$ values will underestimate the AMF compared to the DC3 profile; e.g. a simulation using 250 mol flash$^{-1}$ would likely produce an AMF $\approx 1.3$.

This is further borne out by the profiles in Fig. 5, where it is apparent that increasing the lightning emissions from 0 to 500 to 665 mol flash$^{-1}$ scales the entire profile above $\sim 800$ hPa. Profiles with fewer mol flash$^{-1}$ will have a shape in between the 0 and 500 mol flash$^{-1}$ profiles. Since reducing the UT component of the profile reduces the AMF, these simulations will produce worse agreement with the DC3-derived AMF.

Finally, the 500 mol flash$^{-1}$ nudged run with the base flash rate (approximately half that of the unnudged run, Fig. S3) also demonstrates that lower lightning emissions produces a worse agreement with the DC3-derived AMF, since in the fixed mol flash$^{-1}$ approximation, halving the flash rate is equivalent to halving the mol flash$^{-1}$. Table 2 clearly shows that this run yields an AMF of 1.29, as expected for halving the lightning NO$_2$. Therefore, we do not believe that adding additional runs with lower mol flash$^{-1}$ values would add value to this work.

I am also concerned by the incomprehensive nature of this analysis. The BEHR algorithm is based on several inputs from the operational OMNO2 products, for instance the use of stratospheric NO2 estimates from OMNO2 for calculating tropospheric slant column amount. Modulation of upper tropospheric NO2 should lead to different estimates for stratospheric NO2. Could you do your own stratosphere-troposphere separation and quantify the effect on both stratospheric and tropospheric NO2 estimates?

While we agree that evaluating the impact of using a high spatial resolution model in the stratospheric separation could be interesting, we do not expect the impact to be significant. In Bucsela et al. (2013) (Fig. 2d), most of the continental US (including the southeast where our analysis sees the largest effect of lightning NO$_2$) is already masked during the initial stratospheric separation, since the a priori tropospheric column exceeds the threshold for tropospheric contamination, even at the $2° \times 2.5°$ resolution of the model used. As a quick test, we plotted where our WRF-Chem a priori would exceed the $3 \times 10^{14}$ molec. cm$^{-2}$ threshold for tropospheric contamination from Bucsela et al. (2013) (Fig. R1. The pattern is broadly similar to the masked area in Fig. 2d of Bucsela et al. (2013).

[Figure]

Figure R1: Binary map of the average WRF-Chem tropospheric columns between 1700 and 2200 UTC; red indicates columns greater than the $3 \times 10^{14}$ molec. cm$^{-2}$ threshold for tropospheric contamination from Bucsela et al. (2013).

**Minor comments**

Page 2, lines 27-33: Wouldnt the lifetime of NOx vary with altitude? Suggesting the upper tropospheric NOx lifetime < 4 days might be misleading.

Estimates of NO$_x$ lifetime in the outflow of convection have typically assumed that NO$_x$ lifetime was controlled by dilution and nitric acid formation, giving a lifetime of 2–8 days (Schumann and Huntrieser, 2007). However, more recent work from the DC3 campaign has identified that peroxy radicals formed by reactions of organic precurors lofted from the boundary layer by deep convection lead to rapid formation of alkyl-, peroxy-, and multifunctional- nitrate species in the near field of thunderstorms, indicating that UT NO$_x$ lifetime in the outflow of thunderstorms is shorter than previously assumed (Nault et al., 2016, 2017).

We have added text acknowledging the previous assumption of longer lifetime and explaining in more detail the reason for the shorter lifetime proposed:

> "In the upper troposphere, NO$_x$ lifetime has previously been assumed to be long (2–8 days Schumann and Huntrieser, 2007). Recently, work from the Deep Convective Clouds and Chemistry (DC3) campaign showed that the lifetime of NO$_x$ is short near thunderstorms due to active alkyl-, peroxy-, and multifunctional-nitrate chemistry with peroxy radicals formed in the near field from organic precursors lofted from the boundary layer ($\sim$ 3 h Nault et al., 2016), but longer (12–48 h Nault et al., 2016) away from thunderstorms once these radical species are consumed and other controlling factors take over (Bertram et al., 2007; Apel et al., 2012). In either case, lightning NO$_x$ can affect upper tropospheric NO$_x$ concentrations distant from active storms..."

Page 4, line 17, Section 2.3: What is the altitude range of DC3 measurements? Discuss

how you treat simulated profile beyond the range of DC3 measurements.

DC3 measurements cover the range 980 to 178 hPa. We added the following to Section 2.3:

> "Matching the vertical position in this way inherently restricts the model data to the vertical range of the observations."

And to section 2.5.2:

> "When using the DC3-WRF matched profiles (Sect. 2.3), the two greatest surface pressures (1013 and 989) will have essentially no difference, as the matched profiles only extend down to 990 hPa."

Page 5, line 19: How was the estimate of ghost column made? Does this mean, you add a-priori-derived NO2 columns below the cloud to the retrieved tropospheric columns? How does this approach compare with the operational (DOMINO, OMNO2) procedures?

We have added the following text at p. 5, l. 19:

> "The ghost column was estimated by using as the AMF the ratio of the visible modeled slant column (derived from the a priori $NO_2$ profile, scattering weights, and radiance cloud fraction) to the total modeled tropospheric vertical column. Thus, dividing the observed slant column by this AMF produced a total tropospheric vertical column via a multiplicative correction. This approach is identical to that described in Boersma et al. (2002)."

As mentioned at the end of the above quote, this approach is that described in the OMNO2 Theoretical Basis Document (Boersma et al., 2002). This is indeed the method used in the NASA Standard Product (E. Bucsela, private communication).

Page 5, line 24: What is the logic behind using the fixed tropopause pressure at 200 hPa? Does that mean the BEHR NO2 columns represent columns below 200 hPa? How do you deal with possible errors from using OMNO2-based stratospheric NO2 columns that is likely based on (variable) meteorology-based tropopause pressures?

The 200 hPa upper limit is retained from the original NASA SP v1 approach, where the troposphere profiles end a 200 mbar (Bucsela et al., 2006). Bucsela et al. (2013) indicates that different definitions of the tropopause are expected to have little effect on the retrieval. In any case, since the results presented here focus on differences among AMFs all calculated with 200 hPa as the upper integration limit, much of the effect of the tropopause definition will be canceled out by the difference, since its effects are likely systematic over the time period studied.

Page 6, Eqn 4: Here and everywhere else in the text. This should be tropospheric AMF, not total AMF. Correct or clarify this.

Clarified in text, in Eq. 4 and 5 changed to $AMF_{trop}$.

Page 6, line 16: Use of black-sky albedo instead of Lambert-Equivalent Reflectivity (LER) should be a large source of errors. Please comment on this based on some recent publications (e.g. Lin et al., 2015; Vasilkov et al., 2017).

[Figure]

Figure R2: LER surface reflectance calculating using the SCIATRAN model and the method of Vasilkov et al. (2017) with the MODIS MCD43C1 BRDF product compared against surface reflectance calculated directly from the MODIS MCD43C1 BRDF coefficients and the kernels described in Stahler et al. (1999) for 1 day per month for a year for 85 sites across the continental US. The black dashed line is a reduced major axis regression; the fit indicates only a $\sim 3\%$ difference on average.

The differences found in Lin et al. (2015) and Vasilkov et al. (2017) aren't solely due to the difference between an LER and black-sky albedo; both investigated the difference between a climatological surface reflectance with no directional dependence (e.g. OMLER) and a higher spatial resolution, temporally varying BRDF product that does account for the directional dependence of surface reflectivity.

In response to studies such as Vasilkov et al. (2017), we are already working on switching from a black-sky albedo product to the MODIS BRDF product in the next version of the BEHR retrieval. We examined whether calculating an LER was necessary or if using the surface reflectance computed from the MODIS BRDF parameters was sufficient by computing both for one day from each month for 85 sites across the US, spanning urban, power plant, and rural location. Using the method from Vasilkov et al. (2017), we see only a 3% difference using the LER on average (Fig. R2). While the reviewer is correct to point out the importance of the surface reflectance product, it is not relevant for this paper, as our goal is to isolate the effect of lightning $NO_2$ in a retrieval with high spatial resolution a priori $NO_2$ profiles.

Page 6, line 18-20: Will this approach capture the seasonal variation of surface pressure? How big is its effect on AMF?

From a WRF-Chem simulation of 2012, the average difference in meteorological surface pressure between June and January does not exceed 15 hPa (1.5% of 1013 hPa). From the AMF sensitivity test of the average WRF 500 mol flash$^{-1}$ (unnudged) profile, by randomly sampling the effect of a 15 hPa change in surface pressure at 10,000 points, we find that is

[Figure]

Figure R3: Average difference in surface pressure for all WRF model grid cells between Jan and June 2012.

[Figure]

Figure R4: Domain-wide mean WRF-Chem NO$_2$ profiles. (a) profiles in mixing ratios; (b) profiles in shape factor as defined in Palmer et al. (2001), i.e. number density divided by VCD.

produces only a $\sim 2\%$ average difference in AMF.

Page 6, line 22: OMI cloud fraction.... Is this "effective" or "radiative" cloud fraction? This is the geometric, a.k.a. effective cloud fraction. Corrected.

Page 6-7, Section 2.5.2: Related to Surface pressure (cloudy). I cannot understand the logic of having different surface pressures for clear and cloudy pixels? Should not this be cloud pressure instead? In the calculation of cloudy pixel scattering weights, the cloud pressure is treated as the surface pressure. We have edited this section to make this clearer.

Page 8, Figure 1: Please, also include profile shapes which might be more relevant for AMF. Added (see Fig. R4)

Page 9, lines: 1:3: This discussion is confusing. Should not the effect be based on the altitude of lightning generated NOx?

This discussion is focused on the effect of lesser (higher altitude) surface pressure among the three profiles presented in Fig. 1. Among those three profiles, the altitude of the emitted lightning $NO_2$ does not change; only the amount of lightning $NO_2$ varies. In general, yes, the altitude of lightning emission will change the specific behavior, but in the specific simulations being discussed here, the lightning emissions significantly affect the profile above $\sim 860$ hPa.

Page 10, line 2: In obscure the surface NO2. . ., dont you mean below cloud?

No; a cloud will always, by definition, obscure the below cloud $NO_2$, our point is that clouds are almost always high enough to hide the part of the $NO_2$ profile influenced by surface emissions, and so under cloudy conditions, the entire visible part of our modeled profile is influenced by lightning emissions. We have clarified this as:

> "...the cloud is at sufficiently high elevation to obscure **the part of the $NO_2$ profile influenced by surface emissions**, and therefore restricts..."

Page 16, Section 4.1: What is the message of this section? Why mention nudging at all in the paper if understanding of the 50% decrease in the flash rates by activating FDDA nudging is beyond the scope of the paper?

Laughner et al. (2016) showed that the day-to-day variation in $NO_2$ profiles driven by changes in wind direction has significant effects on the retrieved $NO_2$ and especially on top-down emissions constraints using the exponentially modified gaussian method (Beirle et al., 2011; Valin et al., 2013; Lu et al., 2015; Liu et al., 2016, 2017). Consequently, constraining the WRF meteorology using FDDA nudging is an essential part of producing accurate a priori profiles, and our tests showed that including FDDA nudging also affects the lightning flash rates. Therefore, any model optimization intended to produce high resolution a priori $NO_2$ profiles with lightning emissions should be done with FDDA nudging (or its equivalent in other models) enabled. The discussion with respect to Wong et al. (2013) points out that we do not know that the lower flash rate produced by FDDA nudging is wrong, just that the resultant $NO_2$ profile has poorer agreement than one produced with greater flash rates. We have added the following text to clarify this:

> "A full analysis of the reason that activating FDDA nudging causes the flash rates to decrease by 50% in our case is beyond the scope of this paper. **Empirically, we see that the $NO_2$ profile generated by the FDDA run with 1x the base flash rate has less UT $NO_2$ than was observed during DC3 (Fig. 5). Therefore, we cannot say whether this discrepancy in the profile is due to the reduced number of flashes or a too-low average number of moles of NO emitted per flash. Our correction of doubling the nudged flash rate to improve agreement between the modeled and observed profiles was the most straightforward based on the differences between the nudged and unnudged runs.**"

Page 16, line 22: In less UT NO2 that $=>$ less UT NO2 than

Corrected, thank you.

Page 16, Section 4.2: To estimate uncertainty in cloud slicing, Choi et al. 2014 might have conducted more comprehensive analysis, considering errors in cloud and other parameters. We have added text acknowledging this possibility:

[revised manuscript text omitted]

Vasilkov, A., Qin, W., Krotkov, N., Lamsal, L., Spurr, R., Haffner, D., Joiner, J., Yang, E.-S., and Marchenko, S.: Accounting for the effects of surface BRDF on satellite cloud and trace-gas retrievals: a new approach based on geometry-dependent Lambertian equivalent reflectivity applied to OMI algorithms, Atmos. Meas. Tech., 10, 333–349, doi:10.5194/amt-10-333-2017, URL `https://www.atmos-meas-tech.
[revised manuscript text omitted]